# Expression of miRNAs Targeting ATP Binding Cassette Transporter 1 (ABCA1) among Patients with Significant Carotid Artery Stenosis

**DOI:** 10.3390/biomedicines9080920

**Published:** 2021-07-30

**Authors:** Seonjeong Jeong, Ji Hye Jun, Jae Yeon Kim, Hee Jung Park, Yong-Pil Cho, Gi Jin Kim

**Affiliations:** 1Asan Medical Center, Department of Surgery, University of Ulsan College of Medicine, Seoul 05505, Korea; seonjjeong@gmail.com; 2Department of Biomedical Science, CHA University, Seongnam 13488, Korea; jihyejun1015@gmail.com (J.H.J.); janejykim92@gmail.com (J.Y.K.); heejung970328@gmail.com (H.J.P.); 3Research Institute of Placental Science, CHA University, Seongnam 13488, Korea

**Keywords:** ABCA1, microRNA, carotid stenosis artery, atherosclerosis

## Abstract

**Background:** Carotid artery stenosis is a dynamic process associated with an increased risk of cardiovascular events. However, knowledge of biomarkers useful for identifying and quantifying high-risk carotid plaques associated with the increased incidence of cerebrovascular events is insufficient. Therefore, the objectives of this study were to evaluate the expression of ATP binding cassette transporter 1 (ABCA1) and validate its target microRNA (miRNA) candidates in human carotid stenosis arteries to identify its potential as a biomarker. **Methods:** In human carotid stenosis arterial tissues and plasma, the expression of ABCA1 and its target miRNAs (miRNA-33a-5p, 33b-5p, and 148a-3p) were evaluated by quantitative real time-polymerase chain reaction (qRT-PCR), immunohistochemistry, and enzyme-linked immunosorbent assay (ELISA). **Results:** The expression of ABCA1 was significantly decreased in the plasma of stenosis patients, but its expression was not different in arterial tissues (*p* < 0.05). However, significantly more target miRNAs were secreted by stenosis patients than normal patients (*p* < 0.05). Interestingly, lipotoxicity induced by the oleic and palmitic acid (OAPA) or lipopolysaccharide (LPS) treatment of human umbilical vein endothelial cells (HUVECs) dramatically enhanced the gene expression of adipogenic and inflammatory factors, whereas ABCA1 expression was significantly decreased. **Conclusions:** Therefore, miRNA-33a-5p, 33b-5p, and 148a-3p represent possible biomarkers of carotid artery stenosis by directly targeting ABCA1.

## 1. Introduction

The higher degree of carotid artery stenosis is linked to a higher risk of ipsilateral neurologic events, and significant carotid artery stenosis (greater than 50% diameter reduction) accounts for about 30% of ischemic strokes with an annual risk of ipsilateral stroke of 1–3% [1,2,3]. Carotid artery stenosis—mainly due to carotid atherosclerosis—is a dynamic process associated with an increased risk of cardiovascular events, which persists over time even in cases of asymptomatic mild carotid artery stenosis [4]. Currently, a widely accepted concept is that carotid plaques go through a remodeling process, and sometimes atherosclerosis with even low-grade carotid artery stenosis may result in cerebrovascular events [5,6]. Thus, plaque characteristics other than the degree of stenosis alone may be important for identifying subjects at high risk of stroke [7]. However, there is little knowledge on biomarkers that is useful for identifying and quantifying high-risk carotid plaques associated with the increased incidence of cerebrovascular events.

MicroRNAs (miRNAs) are single-stranded small non-coding RNAs of approximately 19 to 28 nucleotides that participate in regulating the expression of genes through binding to the 3′ untranslated region (3′-UTR) of target messenger RNAs (mRNAs) [8]. miRNAs are important in a variety of biological mechanisms such as differentiation, development, and cell proliferation [9]. miRNA is transcribed by RNA polymerases II or III to primary miRNA with a cap and a poly-A tail. Then, it is processed into short 70-nt stem-loop structures known as pre-miRNAs by a protein complex. Pre-miRNAs are processed into miRNA duplexes by DICER. The miRNA duplex is accompanied by the protein argonaute (AGO) and forms an RNA-induced silencing complex (RISC). This complex targeted mRNA regulates gene expression by translational repression or mRNA degradation [10]. They are processed in the blood as subcellular foci, where some of the mRNAs are stored or de-capped and degraded [11]. miRNAs are secreted from the cell by exosomes and microvesicles [12]. These are found in different types of body fluids such as saliva, serum/plasma, urine, cerebrospinal fluid, and synovial fluid [13]. Therefore, the analysis of miRNA can be used to identify biomarkers for pathophysiological changes and diseases [14].

Atherosclerosis is a chronic inflammatory process of the arterial wall associated with various atherosclerosis risk factors [15]. Higher low-density lipoprotein (LDL) and lower high-density lipoprotein (HDL) levels are considered important contributors to the development of atherosclerosis and cardiovascular disease [16]. The ATP-binding cassette transporter 1 (ABCA1) is an integral cell-membrane protein that regulates the HDL biogenesis by the transport of excess cellular free cholesterol [17]. The interaction of apoA-I with ABCA1 stimulates signal transduction involved in both the anti-inflammatory and lipid efflux mechanisms mediated by ABCA1 [18]. Increased ABCA1 activity induces a significant decline in the size of atherosclerotic lesion, likely due to increased efflux and changes in level and composition of HDL [19]. However, limited studies to date have examined the role of miRNAs in the progression of carotid atherosclerosis and carotid stenosis.

Therefore, the objectives of this study were to evaluate the expression of ABCA1 and validate its target miRNA candidates in human carotid artery stenosis to identify possible biomarkers.

## 2. Materials and Methods

### 2.1. Study Design and Population

This study was performed prospectively with approval from the Institutional Review Board (2019-0873) of Asan Medical Center. All information pertaining to subjects and all human samples were used in compliance with Korean legislation, and all human participants provided written informed consent. This study employed human carotid plaques and blood samples from patients (aged 50–80 years) who had significant carotid bifurcation stenosis (i.e., ≥70% in asymptomatic patients and ≥50% in symptomatic patients) based on the North American Symptomatic Carotid Endarterectomy Trial criteria) on Doppler ultrasound (DUS) imaging and underwent carotid endarterectomy (CEA) (Figure 1 and Appendix A) [2]. Symptomatic carotid disease is defined as neurologic symptoms that are sudden in onset and referable to the appropriate carotid artery distribution (ipsilateral to significant carotid atherosclerotic pathology), including one or more transient ischemic attacks (TIAs) characterized by focal neurologic dysfunction or transient monocular blindness, or one or more minor (non-disabling) ischemic strokes. The definition is contingent on the occurrence of carotid symptoms within the previous six months. To evaluate the specific serum biomarkers of the patients with significant carotid artery stenosis, we compared the serum biomarkers of patients with significant carotid artery stenosis (*n* = 50) with those of normal subjects without carotid artery stenosis or other atherosclerosis risk factors (*n* = 6). The medical information of the control group was taken by history taking from the normal control group, and they have no risk factor for atherosclerosis. The control group was all out-clinic. We selected patients aged 60–70 ages to compare, and mean ages are the same as study groups. Their mean age was 63.8 years (median, 63 years; range, 60–71 years), 33% of the patients were men, and baseline laboratory tests, including lipid battery, were within normal range. The normal subjects also provided written informed consent. The demographic characteristics, risk factors of interest, clinical characteristics, outcomes, DUS findings of carotid diameter reduction, plaque morphology, and plaque area were recorded for all consecutive patients in Excel (Microsoft Corp., Redmond, WA, USA) database and analyzed.

### 2.2. Index Procedures and Tissue Sampling

The surgical procedures were performed as previously detailed [20]. A conventional endarterectomy was performed with patch angioplasty in the standard fashion under general anesthesia and routine carotid shunting. The carotid plaques removed during CEA and the blood samples taken before CEA in 50 patients were analyzed. The internal carotid region was harvested as a disease sample, and the common carotid region was considered a normal sample for comparison with the internal region. Postoperatively, the patients were given dual antiplatelet therapy (low-dose aspirin plus clopidogrel) with a statin in combination with stringent blood pressure control and close observation in an intensive care unit for at least 24 h [21]. All patients were followed up, both clinically and by magnetic resonance imaging with angiography or carotid DUS before discharge.

### 2.3. Clinical Outcomes

The clinical outcomes were defined as a fatal or nonfatal stroke or a transient ischemic attack ipsilateral to the CEA during the perioperative period (within 30 days after CEA). Neurologic events were defined as previously detailed [4,20,22].

### 2.4. Prediction of miRNAs Targeting ATP Binding Cassette Transporter 1 (ABCA1)

The miRNA targets were predicted using a 95% context percentile based on theTargetScan6.2 database (http://www.targetscan.org/vert_71/, accessed on 8 March 2018) [23].

### 2.5. ABCA1 ELISA Analysis

The levels of ABCA1 were analyzed by enzyme-linked immunosorbent assay (ELISA). The concentrations were quantified using the human ABCA1 (MyBioSource, San Diego, CA, USA) ELISA kit in strict accordance with the manufacturer’s instructions and detected in 450nm using a microplate reader (BioTek, Winooski, VT, USA). All reactions were analyzed in triplicate.

### 2.6. Quantitative Real-Time Chain Reaction (qRT-PCR)

Total RNA was isolated from the human carotid stenosis arteries using TRIzol (Invitrogen, Carlsbad, CA, USA). Also, miRNA from plasma was extracted using the miRNeasy Serum/Plasma Kit (Qiagen, Hilden, Germany). Reverse transcription was performed with 500 ng of total RNA and Superscript III reverse transcriptase (Invitrogen). Complementary DNA (cDNA) was amplified by PCR. In the case of cDNA synthesis for miRNAs, we used the miR-X miRNA First-Strand Synthesis kit (Takara Bio, Kusatsu, Shiga, Japan). Real-time PCR was performed using SYBR Master Mix (Roche, Basel, Switzerland) and a CFX Connect™ Real-Time System (Bio-Rad, Hercules, CA, USA). The gene expression was normalized to that of human glyceraldehyde 3-phosphatedehydrogenase (GAPDH) and U6 for miRNA expression. The sequences of the primers are shown in Appendix A. All reactions were performed at least in triplicate.

### 2.7. Immunohistochemistry

To analyze the expression of ABCA1, α-smooth muscle actin (SMA), and CD56 in carotid stenosis artery tissues, we used ABCA1 (Novus, Centennial, CO, USA), α-SMA (Dako, Santa Clara, CA, USA), and CD56 (Abcam, Cambridge, UK) antibodies, respectively. The arterial sections were incubated with 3% H_2_O_2_ in methanol to block endogenous peroxidase activity. After antigen retrieval, the slides were incubated with an anti-ABCA1, α-SMA, CD56 antibodies diluted at 1:100, 1:400, and 1:100 respectively at 4 °C overnight, and then incubated for 1 h with a biotinylated secondary anti-rabbit and mouse antibody at room temperature (RT). The sections were incubated with a horseradish peroxidase-conjugated streptavidin-biotin complex (Dako) and 3,3-diaminobenzidine (EnVision Systems, Santa Clara, CA, USA) was used to visualize the chromatic signals. Images were taken using a digital slide scanner (3DHISTECH Ltd., Budapest, Hungary). Finally, the percentage of positive signals was quantified in all sections by 3DHISTECH Ltd. QuantCenter 2.2 program (3DHISTECH Ltd.).

### 2.8. In Vitro Study with HUVECs

As shown in Figure 8A, to analyze the effects of ABCA1 and its miRNAs in vitro as well as patient samples, human umbilical vein endothelial cells (HUVECs) were cultured in endothelial cell medium (ECM; ScienCell, Carlsbad, CA, USA) under 5% CO_2_ at 37 °C. To induce inflammation and lipotoxicity, oleic acid (OA; 200 μM; Sigma, St. Louis, MO, USA), palmitic acid (PA; 100 μM; Sigma), and lipopolysaccharide (LPS; 20 ng/mL; Sigma) were treated with single or combination for 24 h. Also, 20nM of the inhibitors of each miRNA were transfected using the Lipofectamine 2000 (Invitrogen) for 24 h.

### 2.9. Cloning

The 3′ untranslated regions (UTRs) of the gene sequences targeted by the miRNAs were investigated by Microrna (http://www.microrna.org/microrna/home.do, accessed on 5 June 2021) and used to design oligonucleotides for the luciferase assay. The region of mRNA 3′ UTR was amplified using PCR. The solution of fragment synthesis was contained 10 × Taq buffer (Solgent Co.), 10 mM dNTP mix (Solgent Co.), and h-Taq (Solgent Co.). The synthesis conditions were an initial denaturation at 95 °C for 5 min, 35 cycles of denaturation at 95 °C for 5 s, annealing at 60 °C for 30 s, extension at 72 °C for 1 min, and a pause at 4 °C. The PCR products were purified using a PCR purification kit (Bioneer, Daejeon, Korea). The PCR products were inserted into multiple cloning sites of a pmirGLO vector (Promega, Madison, WI, USA). The linearized pmirGLO vector was mixed with the PCR products and ligase using a ligation protocol. The complete pmirGLO vector was transformed into competent cells and spread onto an ampicillin-containing LB plate. After 24 h of incubation, colonies were selected and cultured in an ampicillin-containing LB broth. Plasmid DNA was extracted using a Plasmid mini extraction kit (Bioneer). The fidelity of the plasmid DNA sequence was confirmed by DNA sequencing.

### 2.10. Dual Luciferase Assay

To analyze ABCA1 3′-UTRs containing the miR-33a-5p, 33b-5p, and 148a-3p binding sites, we constructed a pmirGLO luciferase miRNA target expression vector (Promega, Madison, WI, USA) and verified the results of sequencing (Bioneer, Daejeon, Republic of Korea). Human umbilical vein endothelial cells (HUVECs) were seeded on 24-well plates and transfected with each miRNA mimic or negative control (NC) of 20nM using Lipofectamine 2000 (Invitrogen) for 24 h. The relative luciferase activities were measured by luminescence (Tecan, Switzerland).

### 2.11. Statistical Analysis

All experiments were conducted in duplicate or triplicate. Differences between different regions were analyzed with GraphPad Prism software for comparison between pairs by One-way ANOVA. Datasets with two groups were analyzed with a nonparametric Student’s *t*-test. Significance was taken as * *p* < 0.05.

## 3. Results

### 3.1. Characteristics of Carotid Artery Stenosis

Between July 2019 and April 2020, 50 patients with carotid artery stenosis who underwent CEAs with tissue and blood sampling were consecutively enrolled in this prospective study. The baseline and clinical characteristics of the included patients are presented in Table 1. Their mean age was 68.2 years (median, 67 years; range, 53–79 years), and 88% of the patients were men. The mean degree of carotid artery stenosis was 75.3% (median, 73.5%; range, 54–90%). Among the study population, 13 symptomatic patients (26%) presented with neurological symptoms (within six months prior to CEA) ipsilateral to the carotid artery stenosis; six had strokes, and seven had transient ischemic attacks. All of the 37 asymptomatic patients (74%) presented with carotid artery stenosis with greater than 70% diameter reduction. Preoperatively, most patients (49/50, 98%) had been taking a statin for different durations. Postoperatively, all patients were given a statin, and there were no perioperative neurologic or other cardiovascular events associated with the CEAs.

### 3.2. Selection and Validation of miRNAs Targeting ATP Binding Cassette Transporter 1 (ABCA1)

To determine the miRNAs targeting ABCA1, we searched two miRNA search sites (e.g., TargetScan and miRDB). In the sites, six miRNAs (miR-27a-3p, 27b-3p, 33a-5p, 33b-5p, 144-3p, and 148a-3p) were overlapped as target miRNAs of ABCA1. Among the six miRNAs, the target scores of miR-33a-5p, 33b-5p, and 148a-3p were the highest (Figure 2A). As shown in Figure 2B, miR-33a-5p and 33b-5p bound to two sites of the ABCA1 3′ untranslated region (UTR) region, and miR-148a-3p bound to one site of the ABCA1 3′UTR region (Figure 2B). To determine whether ABCA1 was a direct target of miR-33a-5p, 33b-5p, and 148a-3p, the 3′ UTR reporters for these target genes were cloned into HUVECs. Reporter genes with miRNA mimics were also transfected into HUVECs. Interestingly, when transfected with miRNA mimics, the luciferase activity of the ABCA1 was significantly reduced (Figure 2C–E). These results indicate that these three miRNAs directly targeted ABCA1.

### 3.3. Histopathological Transitions in Carotid Stenosis Arteries

The internal carotid artery exhibited accumulated plaque and shrunken arterial walls compared with the common carotid artery (Figure 3A). To evaluate whether plaque, which is the by-product of inflammation and lipid toxicity, was accumulated in the carotid artery, its area was measured in the common and internal carotid regions. As shown in Figure 3B, the plaque area was significantly increased in the internal carotid artery compared to the common carotid artery (*p* < 0.05, Figure 3B). To evaluate the functioning of the arterial wall in stenotic arteries, Col I and α-SMA were analyzed at the mRNA or protein level. No difference in the mRNA level of Col I was seen between the common and internal carotid artery (Figure 3C). However, the α-SMA expression at the mRNA and protein levels was significantly decreased in the internal carotid artery compared to the common carotid artery due to the decreased thickness of the arterial wall of the internal carotid artery (*p* < 0.05, Figure 3D–F). These data demonstrate that histopathological transitions such as the accumulation of plaque and shrinkage of the arterial wall occur in carotid artery stenosis.

### 3.4. Inflammation in Carotid Artery Stenosis

To investigate the degree of the inflammation in the carotid artery, several inflammatory markers (e.g., interleukin-6, IL-6; interferon-gamma, IFN-γ; tumor necrosis factor-alpha, TNF-α) and interleukin-10 (IL-10), which are known as anti-inflammatory factors, were analyzed at the mRNA level by qRT-PCR. In the carotid arteries, the expression of IL-6 had the tendency to increase in the internal part versus the common part of the carotid arteries (*p* = 0.07, Revised Figure 4A).

The level of IL-10 had a slightly decreased tendency in the internal carotid artery compared to the common carotid artery (*p* = 0.08, Revised Figure 4B). However, the expression of IFN-γ was similar between the common carotid artery and the internal carotid artery (Revised Figure 4C). Also, a slight increase in TNF-α was seen in the internal carotid artery compared to the common carotid artery, although it was not significant (Revised Figure 4D). For histological analysis related to inflammation, the marker of natural killer (NK) cells, CD 56 (neural cell adhesion molecule, NCAM), was analyzed by immunohistochemistry. In the common carotid artery, its expression was barely detected. However, its expression was seen in the peripheral parts of the plaques (Revised Figure 4E). The CD56 positive area in the carotid arteries was increased in the internal carotid artery versus the common carotid artery (*p* < 0.05, Revised Figure 4F). These data indicate that carotid stenosis occurs with inflammation, forming plaques.

### 3.5. Expression of ABCA1 and Its Targeting miRNAs in Carotid Artery Stenosis

To analyze the expression of ABCA1 in the carotid artery tissues, the mRNA and protein levels were evaluated by qRT-PCR and immunohistochemistry. The mRNA level of ABCA1 was not different between the common and internal carotid arteries (Revised Figure 5A). However, the protein expression analyzed by immunohistochemical staining showed a difference between the common and internal regions. ABCA1 was expressed in the cytoplasm of the arterial cells (Revised Figure 5B).

In particular, ABCA1 was highly expressed in the common carotid artery, whereas its expression was lower in the internal region due to injury by stenosis (*p* < 0.05, Revised Figure 5C). Also, its target miRNAs, miR-33a-5p, 33b-5p, and 148a-3p, were analyzed by qRT-PCR. In arterial tissues, the expression of miR-33a-5p and 33b-5p was not different, whereas the level of miR-148a-3p was significantly increased (*p* < 0.05, Revised Figure 5D–F). Also, further analysis was done between asymptomatic and symptomatic patients to evaluate clinical correlation with ABCA1 and its target miRNAs. As the results show, the expression of ABCA1 was slightly decreased in symptomatic patients compared to asymptomatic patients, whereas its target miRNAs were increased. However, their expression patterns were not significant in artery tissues (Appendix A).

Similar to these analyses, ABCA1 and its target miRNAs were evaluated in plasma. The soluble form of ABCA1 was significantly decreased in the plasma of stenosis patients compared to the non-stenosis patients as well as between asymptomatic and symptomatic patients (*p* < 0.05, Revised Figure 6A,B). Also, the expression of the target miRNAs was significantly increased in the stenosis patients versus the non-stenosis patients (*p* < 0.05, Revised Figure 6C,E,G). In further analysis, the expression of ABCA1′s target miRNAs showed a dramatic difference between asymptomatic and symptomatic patients, although the result of miR-33b-5p was not (*p* < 0.05, Revised Figure 6D,F,H). Therefore, it showed an inverse correlation between ABCA1 and miRNAs (Revised Figure 7A–C). These data suggest that ABCA1 was decreased in stenosis, and plaques were formed while its target miRNAs were increased in patient plasma rather than in arterial tissues. Also, this study shows that the increased expression of miRNAs was correlated with carotid stenosis severity and their symptoms.

### 3.6. In Vitro Study in HUVECs

To mimic the carotid stenosis environment, HUVECs were treated with oleic and palmitic acid (OAPA) or lipopolysaccharide (LPS) to induce lipotoxicity and inflammation as shown in Revised Figure 8A. In OAPA or LPS-treated HUVECs, mRNA levels of adipogenic markers (e.g., peroxisome proliferator-activated receptor gamma (PPARγ) and leptin) and inflammatory factors (e.g., IFNγ; nuclear factor kappa-light-chain-enhancer of activated B cells (NF-κB)) was significantly induced compared to non-treated control HUVECs. Particularly, NF-κB were strongly expressed and localized in the nucleus of the HUVECs (*p* < 0.05, Revised Figure 9B,D). In addition, the combination of OAPA and LPS treatment dramatically enhanced their expressions. However, their expression was significantly repressed by transfection with miRNA-33a-5p and 148a-3p inhibitors (*p* < 0.05, Revised Figure 8B–D). The ABCA1 expression of mRNA in HUVECs and the soluble form in the HUVEC culture supernatant was decreased in the OAPA and LPS-treated groups but significantly restored by the transfection of miRNA inhibitors groups (*p* < 0.05, Revised Figure 8E,F). Also, its target miRNAs (miR-33a-5p, 33b-5p, and 148a-3p) showed the reverse tendency as the ABCA1 results (*p* < 0.05, Revised Figure 8G–I).

To examine the vascular function following treatment with OAPA and LPS or transfection with miRNA inhibitors, the tube formation assay was performed in HUVECs by Dil-LDL. The untreated or non-transfected group showed normal tube formation, as shown in Revised Figure 8A. The ability of HUVECs to develop vascular structure was sharply reduced by treatment with OAPA and LPS. However, it was significantly recovered by transfection of the miRNA inhibitors (*p* < 0.05, Revised Figure 9A,C). These results demonstrated that ABCA1 was decreased in damaged HUVECs, whereas the expression of its target miRNAs was increased by directly targeting ABCA1 in the HUVECs. The miRNA inhibitors also recovered the injured vascular function.

## 4. Discussion

Cholesterol metabolism is elaborately regulated at the cellular level, and homeostasis is essential for this process [24]. One of several proteins related to cholesterol homeostasis is ATP Binding Cassette Transporter 1 (ABCA1), a transmembrane protein extensively expressed in many tissues [25]. ABCA1 functions in the first and crucial step of HDL-C formation [26]. Considering that low HDL-C levels are an independent risk factor for coronary heart disease (CHD), genetic variants that are known to increase HDL-C levels would be expected to decrease CHD risk, whereas variants associated with lower HDL-C levels would increase CHD risk [27]. However, high HDL-C levels are not always protective of CHD, and Mendelian randomization studies suggest that there is no causal inverse relationship between HDL-C levels and the risk of CHD [28,29]. Deregulation of miRNAs function may be responsible for several pathophysiological disorders, especially in the cardiovascular system [30,31]. However, the role of miRNAs in carotid-related stroke remains mostly unknown. Previous studies suggest a potential regulatory function for several miRNAs on dysfunctional endothelium, contributing to the evolution of carotid plaque toward growth, instability, and rupture. Maitrias et al. evaluated the expression of seven selected miRNAs in human carotid plaques from a small group of patients [32]. Studies based on larger sample sizes are required to determine the potential use of miRNA as biomarkers in carotid stenosis. In terms of biomarkers, we analyzed miRNAs that are related to carotid plaque and peripheral blood system among patients with carotid stenosis. In our study, the expression and localization of ABCA1 were decreased in patients with atherosclerosis and carotid artery stenosis, as well as their plasma (*p* < 0.05, Revised Figure 5B,C and Figure 6A,B).

Generally, miRNA expression shows unique patterns based on developmental stages, specific diseases, specific organs, events, and cancer types. It also regulates the expression of most genes associated with HDL metabolism, including ABCA1. This implies that miRNAs regulate HDL biogenesis, cellular cholesterol efflux, and HDL-C hepatic uptake, thereby controlling all steps involved in reverse cholesterol transport (RCT) [33]. A previous study showed that miRNA-33, which is located within the gene encoding sterol-regulatory element-binding factor-2 (SREBF-2), modulates the expression of genes involved in cellular cholesterol transport and miRNA-33 appears to regulate both HDL biogenesis in the liver and cellular cholesterol efflux [34]. Cifani et al. evaluated the expression of miRNA-33 in peripheral blood and carotid specimens of patients with symptomatic and asymptomatic carotid artery stenosis. This study showed a different expression of miRNA-33 in peripheral blood and carotid specimens between symptomatic and asymptomatic patients. In the asymptomatic group, increases in miRNA-33 expression were seen in the peripheral blood and arterial specimens [35]. In our data, its expression was evaluated in the plasma and carotid artery (common and internal artery) tissues of patients. The expression was significantly increased in plasma of stenosis patients compared to normal patients (*p* < 0.05, Revised Figure 6C–F). However, in the common and internal carotid artery tissues, its expression was not different (Revised Figure 5D–F).

Considering the role of ABCA1 and miRNA-33 in lipid metabolism, it is not surprising that such miRNA was involved in severe atherosclerotic carotid artery disease [36]. However, the role of miRNA-33 in atherosclerosis is complex, and its effects on inflammatory mediators seem to be paradoxical. The expression of miRNA-33 has been inversely correlated to pro-inflammatory cytokines [37]. Therefore, the inhibition of miRNA-33 promoted macrophage differentiation and pro-inflammatory cytokine release [38], suggesting that miRNA-33 expression can exert pleiotropic action in the different stages of atherosclerotic disease by affecting other risk factors such as obesity, glucose metabolism, and inflammation [39]. This may explain our results. Dyslipidemia represents a key event in plaque formation and growth, whereas inflammation works during all of the phases of atherosclerosis and represents the main factor leading to plaque instability [40].

The pro-inflammatory factors, protein mediators, and other factors such as free fatty acids will aggravate the damage of endothelial cells and then induce or aggravate the process of atherosclerosis [41,42]. NF-κB, associated with innate and adaptive immunity as well as in inflammatory diseases, regulates the pro-inflammatory cytokines and NO release [43]. Studies have confirmed that NF-κB regulates the development of atherosclerotic disease [44]. Consistent with this explanation, we confirmed the markers related to dyslipidemia (e.g., PPARγ and leptin) in oleic and palmitic acid (OAPA) and lipopolysaccharide (LPS) treated-HUVECs for mimicking atherosclerosis of artery as well as ABCA1 and its target miRNAs. The expression of PPARγ and leptin, as well as the expression of target miRNA of ABCA1, was dramatically increased in HUVECs damaged by treatment with OAPA and LPS (*p* < 0.05, Revised Figure 8B,C,G–I). Also, ABCA1 was repressed by the treatment of HUVECs with OAPA and LPS and restored by miRNA inhibitors (*p* < 0.05, Revised Figure 8E,F). The use of statin and its effects on the lipid profile is known to be associated with a significant decrease of miRNA-33 and -148a expression, and in recent years, clinically, statin has been used extensively in patients with atherosclerosis risk factors [45,46]. In our study, most patients (49/50) had been taking statin preoperatively for different durations, and all patients were given a statin after CEA. This could impose bias in our study and is an important limitation that should be acknowledged. Also, our current findings were obtained at a single center, resulting in a small sample size, which limits the overall relevance of our results, and some clinical information in normal subjects was not available from the medical records. The study cohort was entirely Asian; therefore, these results may not be generalizable to other ethnic groups. Further studies are needed for more robust clinical correlation evidence of target miRNAs.

## 5. Conclusions

In carotid artery stenosis, histopathological transitions such as the accumulation of plaque and shrinkage of the arterial wall occurred as well as inflammatory reactions (Figure 10). ATP Binding Cassette Transporter 1 (ABCA1) was decreased by stenosis and the formation of plaques while its target miRNAs (miRNA-33a-5p, 33b-5p, and 148a-3p) were significantly increased in patient plasma but not in arterial tissues. Therefore, we suggest the potential of miRNA-33a-5p, 33b-5p, and 148a-3p, which directly target ABCA1, as biomarkers of carotid artery stenosis.

## Figures and Tables

**Figure 1 biomedicines-09-00920-f001:**
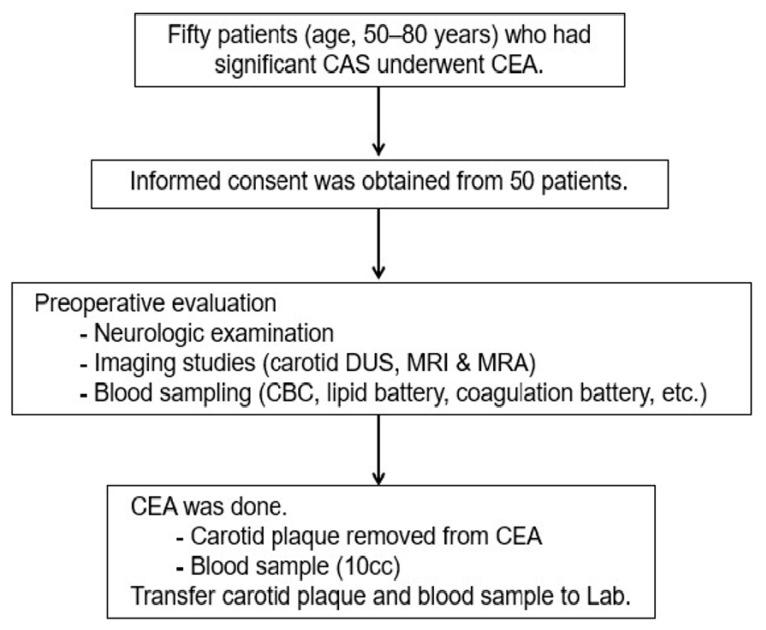
Diagram of the study design. CAS, carotid artery stenosis; CEA, carotid endarterectomy; CBC, complete blood count; DUS, Doppler ultrasound; MRA, magnetic resonance angiography, MRI, magnetic resonance imaging.

**Figure 2 biomedicines-09-00920-f002:**
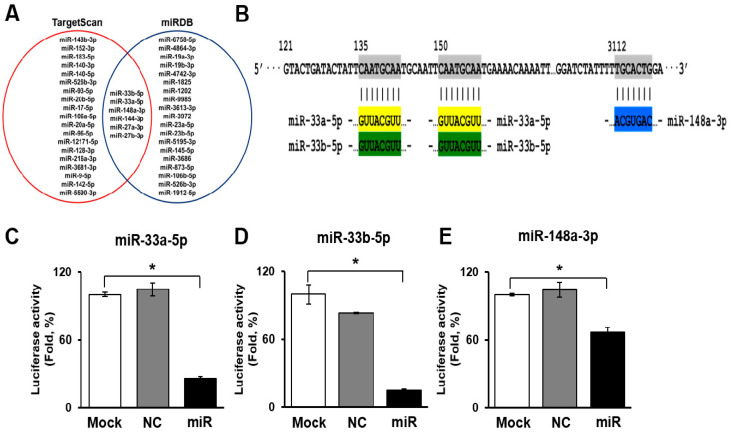
miRNAs targeting ABCA1. (**A**) Venn diagram of miRNAs targeting ABCA1. (**B**) The 3′ UTR of the ABCA1 gene containing miRNA binding sites for miR 33a-5p, miR-33b-5p, and miR-148a-3p. Luciferase assays to validate the interaction of miR-33a-5p (**C**), miR-33b-5p (**D**), and miR-148a-3p (**E**) with ABCA1. Mean ± SD (*n* = 3 per group); * *p* < 0.05; *t*-test. NC, negative control; miR, miR transfected group.

**Figure 3 biomedicines-09-00920-f003:**
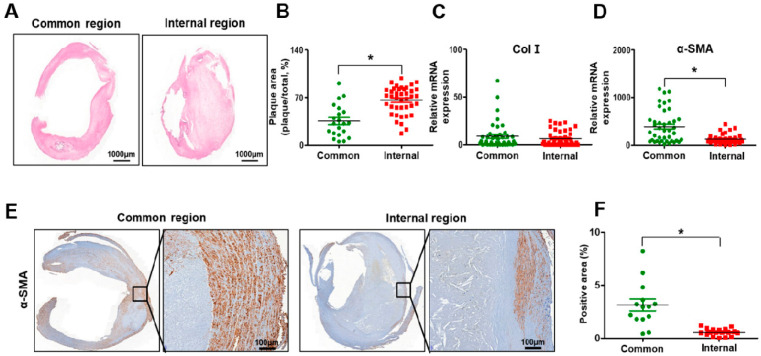
Histopathological transition in the carotid stenosis artery. (**A**) Hematoxylin and eosin staining in the common and internal carotid artery. (**B**) Quantification of plaque area by hematoxylin and eosin staining. mRNA expression of collagen I (**C**) and alpha-smooth muscle actin (**D**) by qRT-PCR. (**E**) Expression of alpha-smooth muscle actin by immunohistochemical staining. Scale bar: 100 μm. (**F**) Quantification of the alpha-smooth muscle actin positive area by immunohistochemistry staining. Mean ± SD; * *p* < 0.05; *t*-test. Common, Common carotid artery; Internal, Internal carotid artery; Col I, collagen I; α-SMA, alpha-smooth muscle actin.

**Figure 4 biomedicines-09-00920-f004:**
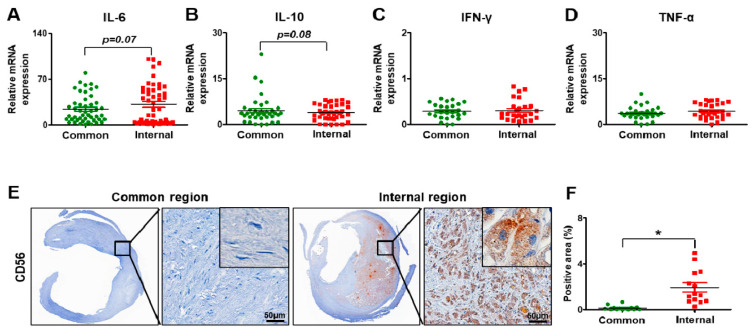
Inflammation in carotid artery stenosis. mRNA expression of interleukin-6 (**A**), interleukin-10 (**B**), interferon-gamma (**C**), and tumor necrosis factor-alpha; (**D**) by qRT-PCR. (**E**) Expression of CD56 by immunohistochemical staining. Scale bar: 50 μm. (**F**) Quantification of the CD56 positive area. Mean ± SD; * *p* < 0.05; *t*-test. Common, common carotid artery; Internal, internal carotid artery; IL-6, interleukin-6; IL-10, interleukin-10; IFN-γ, interferon-gamma; TNF-α, tumor necrosis factor-alpha.

**Figure 5 biomedicines-09-00920-f005:**
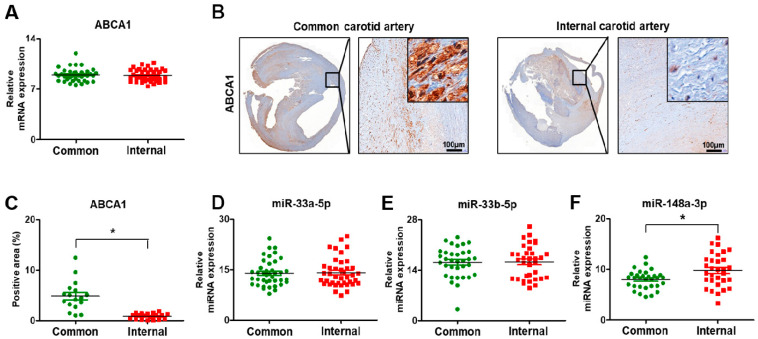
Expression of ABCA1 and its target miRNAs in carotid artery stenosis. (**A**) mRNA expression of ABCA1 by qRT-PCR. (**B**) Expression of ABCA1 by immunohistochemical staining. Scale bar: 100 μm. (**C**) Quantification of the ABCA1 positive area by immunohistochemistry staining. Expression of miR-33a-5p (**D**), miR-33b-5p (**E**), and miR-148a-3p (**F**) by qRT-PCR. Mean ± SD; * *p* < 0.05; *t*-test. Common, common carotid artery; Internal, internal carotid artery.

**Figure 6 biomedicines-09-00920-f006:**
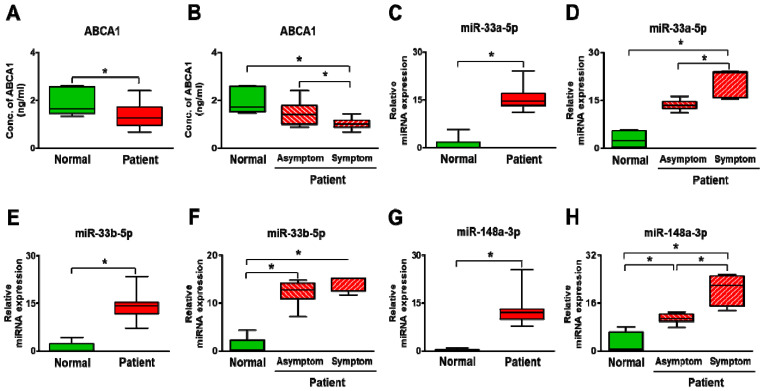
Expression of ABCA1 and its target miRNAs in the plasma of carotid artery stenosis patients. (**A**,**B**) Expression of soluble ABCA1 by ELISA. Expression of miR-33a-5p (**C**,**D**), miR-33b-5p (**E**,**F**), and miR-148a-3p (**G**,**H**) by qRT-PCR. Mean ± SD; * *p* < 0.05; *t*-test or One-way ANOVA. Normal, plasma of non-carotid artery stenosis patient; Patient, plasma of carotid artery stenosis patient.

**Figure 7 biomedicines-09-00920-f007:**
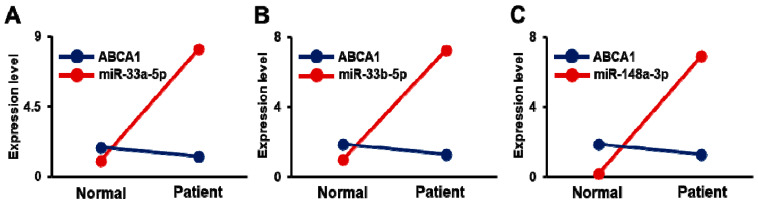
Correlation of ABCA1 and its target miRNAs in the plasma of carotid artery stenosis patients. Correlation between soluble ABCA1 and (**A**) miR-33a-5p, (**B**) miR-33b-5p, and (**C**) miR-148a-3p. Normal, plasma of non-carotid artery stenosis patient; Patient, plasma of carotid artery stenosis patient.

**Figure 8 biomedicines-09-00920-f008:**
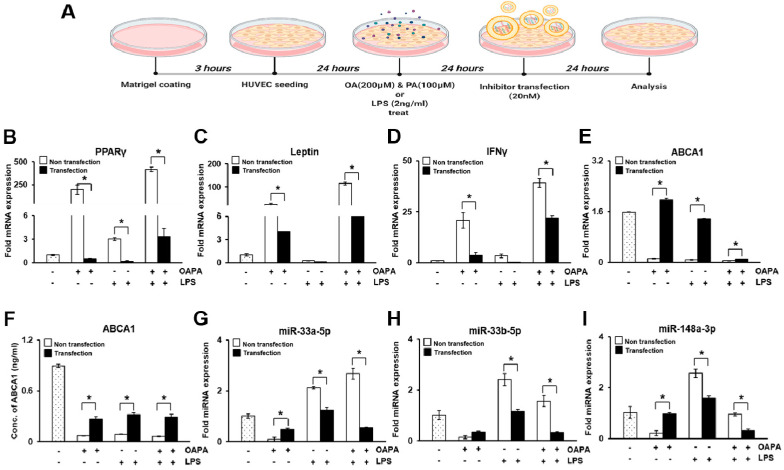
In vitro study in HUVECs. (**A**) Experimental scheme of in vitro experiment using HUVECs. mRNA expression of PPARγ (**B**), leptin (**C**), IFN-γ (**D**), and ABCA1 (**E**) by qRT-PCR. (**F**) Expression of soluble ABCA1 in culture supernatant by ELISA. miRNA expression of miR-33a-5p (**G**), miR-33b-5p (**H**), and miR-148a-3p (**I**) by qRT-PCR. Mean ± SD (*n* = 3 per group); * *p* < 0.05; *t*-test. OAPA, oleic acid, and palmitic acid; LPS, lipopolysaccharide; -, non-treated group; +, treated group.

**Figure 9 biomedicines-09-00920-f009:**
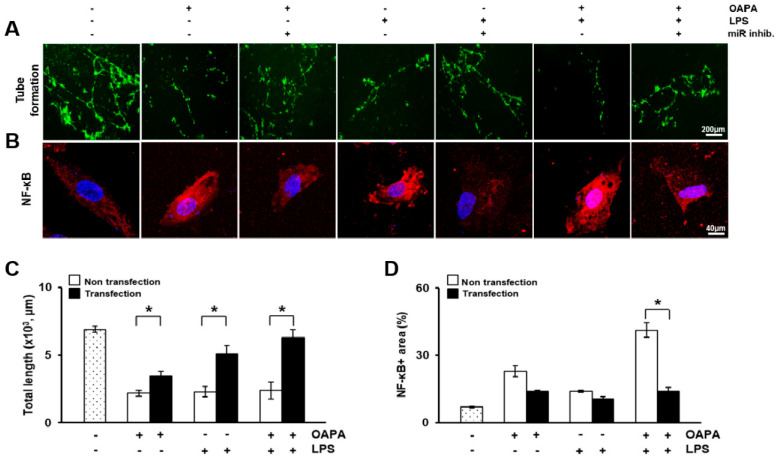
Functional recovery of damaged HUVECs by miRNA inhibitors. (**A**) Tube formation assay in damaged HUVECs by Dil-LDL staining. Scale bar: 200 μm. (**B**) Expression of NF-κB in damaged HUVECs by immunofluorescence. Scale bar: 40 μm. (**C**) Quantification of total tube length by the tube formation assay. (**D**) Quantification of the NF-κB positive area by immunofluorescence. Mean ± SD (*n* = 3 per group). * *p* < 0.05; *t*-test. OAPA, oleic acid and palmitic acid-; LPS, lipopolysaccharide; miR inhib., miR-33a-5p and 148a-3p inhibitor; -, non-treated group; +, treated group.

**Figure 10 biomedicines-09-00920-f010:**
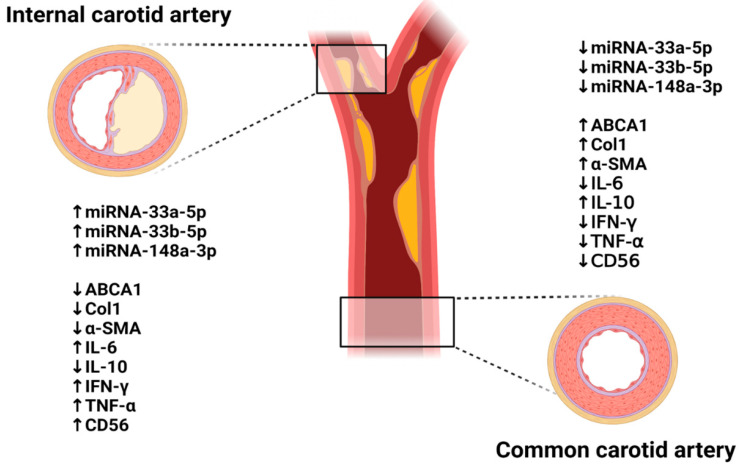
Summary diagram of the global hypothesis in this study.

**Table 1 biomedicines-09-00920-t001:** Baseline demographic and clinical characteristics of the study population.

	Total (N = 50)
Mean age, years	68.2 ± 6.3
Male	44 (88)
BMI, kg/m^2^	24.0 ± 3.2
Risk factors	
Hypertension	32 (64)
Diabetes mellitus	25 (50)
CVD	20 (40)
Heart failure	10 (20)
CKD	5 (10)
Smoking	32 (64)
BMI (kg/m2)	24.2 ± 3.2
Lipid battery *	
LDL (mg/dL)	76.2 ± 25.6
HDL (mg/dL)	43.7 ± 10.6
Triglyceride (mg/dL)	126.54 ± 55.7
Total cholesterol (mg/dL)	129.9 ± 28.0
Use of statin ^†^	49 (98)
Carotid stenosis	
Degree of stenosis, %	
ICA	75.3 ± 7.9
ECA	27.9 ± 12.5
Symptomatic	13 (26)
Stroke	6 (12)
TIA	7 (14)

The continuous data are expressed as mean ± standard deviation and the categorical data as *n* (%). BMI, body mass index; CVD, cardiovascular disease; CKD, chronic kidney disease; ECA, external carotid artery; HDL, high-density lipoprotein; ICA, internal carotid artery; LDL, low-density lipoprotein; TIA, transient ischemic attack. * Preoperative lipid levels, ^†^ Preoperative use of statin.

## Data Availability

Not applicable.

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
