# Peer review of "Expression of miRNAs Targeting ATP Binding Cassette Transporter 1 (ABCA1) among Patients with Significant Carotid Artery Stenosis"

_biomedicines, 2021, doi:10.3390/biomedicines9080920_

Round 1
Reviewer 1 Report
In this interesting study, Jeong et al aimed to identify novel biomarkers of carotid artery stenosis by evaluating the expression of ABCA1 and its target miRNAs in human arterial tissues and plasma. Overall, the idea behind this study is good in terms of novelty and clinical relevance. The manuscript is well written, with a clear presentation of the results. I only have some minor comments.
- The Authors reported a short unstructured Abstract, with no bolded headings. Considering that their manuscript is not a review, a structured and more comprehensive abstract is needed, with a clearer description of their interesting results and reporting of conclusions.
- Please give the acronyms in full at first mention (e.g., “ABCA1” in the Abstract section).
- In the Introduction section, the first 3 sentences (lines 32-40) seem redundant, with each statement appearing as a repetition of the previous one. Please rephrase.
- In Table 1, please provide the rate of obesity, hypercholesterolemia, hypertriglyceridemia and total cholesterol levels.
Author Response
Dear Reviewer,
We greatly appreciate your careful evaluation of our manuscript (Biomedicines-1273825) entitled: “Expression of miRNAs targeting ABCA1 among patients with significant carotid artery stenosis” We were really encouraged by the reviewers’ positive comments and constructive suggestions. I am happy to report that we have successfully addressed all issues and concerns through additional data and subsequent revision of our manuscript, as detailed in the following response page. Changes are highlighted in red in the revised manuscript,.
In summary, based on the insightful and constructive criticisms provided by both referees, we believe that our manuscript is significantly improved and we hope that you will consider it suitable for publication in Biomedicines.
The material contained herein has not been published previously by any of the authors, and is not under consideration for publication in another journal at this time.
Very sincerely yours,
Gi Jin Kim, Ph.D.
Associate Professor
Department of Biomedical Science, CHA University
689, Sampyeong-dong, Bundang-gu, Seongnam-si, Gyeonggi-do, Republic of Korea.
Tel: 82-31-881-3687, Fax: 82-31-881-4102, e-mail: [email protected]

Reviewer 2 Report
This study aimed to evaluate the expression of ABCA1 and its target miRNAs (miR33 and miR148) in human carotid artery stenosis to identify biomarkers. This is a resubmission of an interesting paper that is well and clearly written. However, I have some concerns about the experiments performed:
1- Authors observed that a-SMA expression was decreased in the internal carotid artery suggesting a role of VSMCs but they performed in vitro experiments in endothelial cells (HUVEC). Authors have to justify the in vitro model because it is not derived by the in vivo results, particularly because ABCA1 staining of the endothelium is not clearly seen on IHC.
2-The in vitro study in HUVECs was not described in MM, nor was the use of miRNA inhibitors.
3-I also have difficulty understanding the results in figure E. It seems that cells transfected with miRNA inhibitors regain ABCA1 expression when treated with OAPA or LPS alone, but the effect is attenuated when both stimuli are used (OAPA + LPS ). This effect is intriguing because the levels in the supernatant are stable. This result must be discussed.
3-In IHC, only CD56 was analyzed. Why not other inflammatory cells like macrophages or dendritic cells? Furthermore, the role of NFKb is considered important in the discussion, but it has not been evaluated in arteries, only in the in vitro experiments (Figure 9 B and D). Therefore, the analysis of NFkb activation and other inflammatory cells should be also analyzed in the arteries.
4-Circulating levels of mRNA in the patients should be analyzed.
5-A summary graph of the global hypothesis will be appreciated.
Author Response

(The authors gave the same response as above.)

Reviewer 3 Report
In their response to my previous comment, the authors have done little to revise the areas that I suggested to be changed. Although they agreed with my comments, they did not add much additional content. For this reason, it is my view that this paper should not be published.
Author Response
Dear Reviewer,
We greatly appreciate your careful evaluation of our manuscript (Biomedicines-1273825) entitled: “Expression of miRNAs targeting ABCA1 among patients with significant carotid artery stenosis” We were really encouraged by the reviewers’ positive comments and constructive suggestions. I am happy to report that we have successfully addressed all issues and concerns through additional data and subsequent revision of our manuscript, as detailed in the following response page. Changes are highlighted in red in the revised manuscript,.
In summary, based on the insightful and constructive criticisms provided by both referees, we believe that our manuscript is significantly improved and we hope that you will consider it suitable for publication in Biomedicines.
The material contained herein has not been published previously by any of the authors, and is not under consideration for publication in another journal at this time.
Very sincerely yours,
Gi Jin Kim, Ph.D.
Associate Professor
Department of Biomedical Science, CHA University
689, Sampyeong-dong, Bundang-gu, Seongnam-si, Gyeonggi-do, Republic of Korea.
Tel: 82-31-881-3687, Fax: 82-31-881-4102, e-mail: [email protected]

This manuscript is a resubmission of an earlier submission. The following is a list of the peer review reports and author responses from that submission.
Round 1
Reviewer 1 Report
This study evaluated the expression of ABCA1 and validated its target miRNA candidates in human carotid artery stenosis to identify its potential as a biomarker. The results suggested that miRNA-33a-5p, 33b-5p, and 148a-3p are biomarkers of arterial stenosis.
However, the relationship between mechanisms that increase miRNA and arterial stenosis rates remains unclear. In order to use these miRNAs as biomarkers, the relationship between carotid stenosis arteries and miRNAs should be demonstrated in detail. For the above reasons, it is my view that this paper is not fit for publication.
Major comment:
- In order to use these miRNAs as biomarkers in arterial stenosis, the relationship between arterial stenosis and the miRNAs should be clarified by showing why the expression levels of miRNA-33a-5p, 33b-5p, and 148a-3p are increased.
- It is unclear whether the expression levels of these miRNAs rise early in arterial stenosis or after the plaques are complete. In order to use these miRNAs as biomarkers for arterial stenosis, the relationship between these miRNAs and the stenosis rate should be demonstrated in detail.
Minor comment:
- The arrangement of the figures in the text is disjointed. For example, Fig 5A → Fig 5 E → Fig 5 F → Fig 5 B-D. Therefore, the sentences appearing in the text should be rearranged or the order of the figures should be changed to Fig 5A → Fig 5B-D → Fig 5E → Fig 5F.
Author Response
Dear Reviewer,
We greatly appreciate your careful evaluation of our manuscript (Biomedicines-1186146) entitled: “Expression of miRNAs targeting ABCA1 among patients with significant carotid artery stenosis” We were really encouraged by the reviewers’ positive comments and constructive suggestions. I am happy to report that we have successfully addressed all issues and concerns through additional data and subsequent revision of our manuscript, as detailed in the following response page. Changes are highlighted in red in the revised manuscript,.
In summary, based on the insightful and constructive criticisms provided by both referees, we believe that our manuscript is significantly improved and we hope that you will consider it suitable for publication in Biomedicines.
The material contained herein has not been published previously by any of the authors, and is not under consideration for publication in another journal at this time.
Very sincerely yours,
Gi Jin Kim, Ph.D.
Associate Professor
Department of Biomedical Science, CHA University
689, Sampyeong-dong, Bundang-gu, Seongnam-si, Gyeonggi-do, Republic of Korea.
Tel: 82-31-881-3687, Fax: 82-31-881-4102, e-mail: [email protected]

Reviewer 2 Report
This study aimed to evaluate the expression of ABCA1 and its target miRNAs in human carotid artery bifurcation stenosis in tissue and serum to identify biomarkers. The paper is interesting and it is well and clearly written. However, I have some concerns:
1- A brief summary of the surgical procedure should be added as supplementary material. A video of the procedure would be appreciated.
2- I'm confused about where the normal "common carotid region" came from. Where was its original place? How was it obtained? A drawing showing the origin of the injured area and the normal sample would be appreciated.
3- Clinical characteristics of normal population of subjects without carotid artery stenosis should be also described.
4- Table 1 with 2 columns is confusing. I recommend using only one column to describe the population.
5- In transfection methods, the concentration of mimics and the 3'UTR clone used must be included.
6- The panels in the Figures are too small and the view is not clear enough. Also, the magnification used in Figure 3E does not appear the same (the nuclei appear smaller on the right, although the size bar is the same).
7- The authors have to explain how the percentage of the positive cell was quantified in the IHC.
8- In line 256 the authors say "The level of IL10 was slightly inhibited in the internal carotid region compared to the common carotid region" ... but figure 4B suggests that the mean is displaced by an extreme value and the trend was in the opposite direction. Therefore, these data should be re-analyzed and the results explained with caution.
9- In panel 4E I cannot clearly see the cells that are stained with CD56. I suggest you show a magnified view of cells CD56 +.
10- In line 272, it is said that "ABCA1 was expressed in the cytoplasm of arterial cells." It would be interesting if the authors could reveal which these cells are (I suggest performing a double IHC).
11-Line 285. “The soluble form of ABCA1 was significantly decreased in the plasma of stenotic patients compared to the non-stenotic patients”. It was claimed that there were 10 control patients, but In figure 6A I can only count 6. What happens with the other 4?.
12- Line 288. ABC was decreased in stenosis while miRNAs were increased in patient plasma. However, there are no differences in mRNA expression in tissue. This topic has to be discussed in more detail. Is there an inverse correlation between ABCA1 and miRNAs?. Please show the graph
13- Figures 8A and B have to be magnified and show with arrows what readers have to observe.
14- Why were HUVECs used for the in vitro study? Staining of the endothelium is not clearly seen on ABCA1 IHC.
15- Other limitations of the study have to be discussed, not only statins.
16- Use nonparametric tests in statistics, not just a t-test, for non-normal variables.
Author Response

(The authors gave the same response as above.)

Round 2
Reviewer 1 Report
In their response to my previous comment, the authors have done little to revise the areas that I suggested to be changed. Although they agreed with my comments, they did not add much additional content. For this reason, it is my view that this paper should not be published.
Reviewer 2 Report
I agree with the answers, the manuscript has been improved and is now suitable for publication.